Cyclocytidine hydrochloride inhibits the synthesis of relaxed circular DNA of hepatitis B virus

Wang Xue 1
Xiao Yihan 1
Cui Zhigang 1
Li Zongxin 1
Li Lihua 1
Wu Lixian 1
Yin Feifei yinfeifeiff@163.com 1 2 3 4
Cui Xiuji cuixj26@hainmc.edu.cn 1 2 3 4
1 School of Basic Medicine and Life Science, Hainan Medical University , Haikou , China
2 Key Laboratory of Tropical Translational Medicine of Ministry of Education, Hainan Medical University , Haikou , China
3 NHC Key Laboratory of Control of Tropical Diseases, School of Tropical Medicine, Hainan Medical University , Haikou , China
4 Hainan Medical University—The University of Hong Kong Joint Laboratory of Tropical Infectious Diseases, Hainan Medical University , Haikou , China
Franco Bernardo
Electronic publication date: 2022 Jul 12
Publication date: 2022
Volume: 10
Electronic Location ID: e13719
Received 2022 Apr 11; Accepted 2022 Jun 22
Copyright: ©2022 Wang et al.
Copyright year: 2022
Copyright holder: Wang et al.
License: This is an open access article distributed under the terms of the Creative Commons Attribution License, which permits unrestricted use, distribution, reproduction and adaptation in any medium and for any purpose provided that it is properly attributed. For attribution, the original author(s), title, publication source (PeerJ) and either DOI or URL of the article must be cited.
License URL: https://creativecommons.org/licenses/by/4.0/

Keywords: Hepatitis B virus, Cyclocytidine hydrochloride, Relaxed circular DNA, Chronic hepatitis B

Funding: Hainan Provincial Natural Science Foundation of China No. 820RC640 National Natural Science Foundation of China No. 82060365 81660335 Major Science and Technology Program of Hainan Province ZDKJ202003 Research project of Hainan academician innovation platform YSPTZX202004 This work was supported by the Hainan Provincial Natural Science Foundation of China (No. 820RC640), the National Natural Science Foundation of China (No. 82060365 and 81660335), the Major Science and Technology Program of Hainan Province (No. ZDKJ202003), and the Research project of Hainan academician innovation platform (No. YSPTZX202004). The funders had no role in study design, data collection and analysis, decision to publish, or preparation of the manuscript.

==============================
Background

Cyclocytidine hydrochloride (HCl) has been reported to inhibit DNA synthesis by affecting DNA polymerase. Here, we tested the antiviral effect of cyclocytidine on hepatitis B virus (HBV) DNA synthesis, which is reliant on DNA polymerase activity.

Materials and Methods

Cyclocytidine HCl was treated to HBV-producing HepAD38 cells or added to an endogenous polymerase reaction, and HBV DNA was detected by Southern blot.

Results

Treatment of 20 µM cyclocytidine HCl significantly decreased the production of relaxed circular (rc) DNA in HepAD38 cells and block rcDNA synthesis in endogenous polymerase reaction (EPR), a cell free assay, possibly by inhibiting the HBV DNA polymerase activity.

Conclusion

Cyclocytidine HCl could inhibit the synthesis of HBV rcDNA, the precursor of covalently closed circular DNA, and this result provides a case for the usage of “old” drugs for “new” applications.

Introduction

Hepatitis B virus (HBV) infection can cause hepatitis, leading to chronic hepatitis, liver cirrhosis, and hepatocellular carcinoma (Yuen & Lai, 2000).

HBV is an enveloped DNA virus that belongs to the Hepadnaviridae family, and the virion contains a relaxed circular (rc) DNA, approximately 3.2 kilobases (kb), within the capsid (Delius et al., 1983; Summers & Mason, 1982). rcDNA is a partially double-stranded circular DNA consisting of a negative strand and a positive strand of DNA. The negative strand DNA, denoted nsDNA, has a full-length HBV DNA sequence, whereas the positive strand consists of various sizes of DNA sequences and is shorter than the negative strand. HBV polymerase (POL) possessing DNA polymerase and reverse transcriptase (RT) activity is attached to the 5′ end of negative strand DNA (Summers & Mason, 1982).

Upon entry into hepatocytes, rcDNA released from nucleocapsid (NC) is transported to the nucleus to form covalently closed circular (ccc) DNA by completing the positive strand and ligating gaps on positive or negative stranded DNA, respectively (Tuttleman, Pourcel & Summers, 1986). cccDNA is used as a template for all HBV viral RNAs including pregenomic (pg) RNA that translates core proteins and HBV POL, as well as serving as a template for rcDNA synthesis within NC (Gao & Hu, 2007; Guo et al., 2007; Wu et al., 1990). In the cytoplasm, after core proteins encapsidate pgRNA attached with HBV POL to form NC, HBV initiates the synthesis of the nsDNA using pgRNA as a template using the help of the RT activity of HBV POL. Next, the positive strand of rcDNA was synthesized using the nsDNA as a template by DNA polymerase activity of HBV POL to form rcDNA (Hu & Seeger, 2015; Kaplan et al., 1973). Current anti-HBV agents are mostly nucleos(t)ide analogous that can efficiently inhibit the synthesis of nsDNA or rcDNA through competing with normal nucleotides to terminate the viral DNA extension. However, they will not be able to inhibit or destroy the formation of cccDNA, which is the molecular basis of persistent HBV infection (Hu & Seeger, 2015; Litwin et al., 2005). Thus, it would be an important and valuable work to seek new anti-HBV agent that block cccDNA formation.

Mechanism of cccDNA formation is unclear. However, it had been shown that certain human DNA polymerases, such as polymerase kappa, involved in the cccDNA formation (Qi et al., 2016). In our work routine in screening of bioactive compounds that possess inhibitory activity of human DNA polymerase, we identified cyclocytidine hydrochloride (HCl) as a potent and selective inhibitor of HBV replication, especially inhibit rcDNA synthesis. Cyclocytidine HCl is a prodrug of cytarabine, which is used in treatment of acute myeloid leukemia more than three decades. Cyclocytidine HCl is structurally similar to deoxycytidine that is incorporated into human DNA and also displays the ability to inhibit human DNA polymerase to kill the cancer cells (Gray et al., 1972; Kirsch & Notari, 1984; Lowenberg et al., 2011).

In the present study, cyclocytidine HCl showed an inhibitory effect on the synthesis of rcDNA, the precursor of cccDNA, and the possible mechanisms of inhibition are described below.

Material and Methods

Cell culture and cyclocytidine HCl treatment

HepAD38 cells (kindly provided by Dr. Christoph Seeger from Fox Chase Cancer Center, Philadelphia, PA, USA) that contain an HBV 1.1-fold genome and express wildtype HBV particles under regulation of the tetracycline (Tet) response promoter (Ladner et al., 1997) were cultured in a six cm-cell culture dish containing 1 µg/mL Tet (Cat#S4490; Selleckchem, Houston, TX, USA), 400 µg/mL G418 (Cat#11811031; Thermo Fisher, Waltham, MA, USA), and 50 µg/mL of penicillin/streptomycin in Dulbecco’s modified eagle’s medium (DMEM) (Cat#21013024; Gibco, Waltham, MA, USA) supplemented with 10% fetal bovine serum (Cat#C04001; VivaCell, Shanghai, China). To induce the production of HBV DNA, Tet was removed from the culture medium when cells reached 80% confluence and 5, 10, and 20 µM of cyclocytidine HCl (>99% of purity) purchased from Selleckchem (Cat# s1973) was added to cells at day 7 post-induction for 2 days, respectively. The same volume of dimethyl sulfoxide (DMSO) was added to cells as a mock treatment. After two or four days of cyclocytidine HCl treatment, the level of cytoplasmic HBV DNA expression was detected by a Southern blot.

MTT assay

The cytotoxicity of cyclocytidine HCl on cells was measured using a classical MTT cell viability assay. In brief, HepAD38 cells were cultured in 6-cm culture dish with different amounts of cyclocytidine HCl for 2 days or 4 days. Cells were trypsinized and treated with trypan-blue solution. Then, cells were counted using hemocytometer and were seeded into a 96-well plate (5  × 103 cells/well). Cells were subsequently maintained in serum-free DMEM media for 4 h at 37 °C. After incubation, the culture medium was discarded and 40 mL of MTT solution (5 mg/mL) (Cat#1334GR001; neoFroxxEinhausen, Germany) was added to each well. Next, the 96-well plate was wrapped with foil and shaken on an orbital shaker for 15 min at room temperature. The absorbances of each well was measured at a wavelength 570 nm.

Detection of cytoplasmic HBV DNA

Cytoplasmic HBV DNA within NC, denoted as core DNA, was released by treatment of sodium dodecyl sulfate (SDS) (Sigma-Aldrich, Cat#L3771) and proteinase K (PK) (Cat#25530049; Thermo Fisher) (Cui et al., 2013). Briefly, the cells were lysed with 200 µL of nonidet P (NP)-40 (Cat#18896; Sigma-Aldrich, St. Louis, MO, USA) lysis buffer and centrifuged at 14,000 rpm for 5 min to remove nuclear pellets. A total of 18 µL supernatant was mixed with 10 mM ethylene diamine tetraacetic acid (EDTA), 0.5% SDS, and 0.5 mg/mL PK, and then incubated for 2 h at 37 °C. The released core DNAs were resolved on 1.2% agarose gel and were subsequently transferred and fixed onto a Nylon membrane (Millipore, Cat#INYC00010). The blotted DNAs were hybridized with a digoxigenin-11-dUTP-labeled HBV-specific DNA probes generated with random primed labeling technique using full-length HBV DNA as a template according to the manufacture’s instruction (#11585614910; Roche, Basel, Switzerland). Generated DNA probe was able to hybridize to every region of HBV DNA. Probes hybridized to HBV DNA were immunodetected with anti-digoxigenin-alkaline phosphatase and were then visualized with chemiluminescence substrate CSPD, which can be dephosphorylated by alkaline phosphatase to emit light at a maximum wavelength of 477 nm. The chemiluminescent signal was obtained by Tanon 5200 chemiluminescent imaging system, and the density of the chemiluminescent signal was measured using ImageJ software, https://imagej.nih.gov/ij/index.html.

Endogenous polymerase reaction (EPR)

EPR that allows viral RT to synthesize HBV DNA within NC using endogenous viral RNA and DNA templates were carried out as previously described (Cui et al., 2013; Nguyen, Gummuluru & Hu, 2007). In brief, 20 µL of the cell lysates that contained cytoplasmic NCs were incubated with a 100 mM dNTP mix (dATP, dTTP, dCTP and dGTP) in EPR buffer (50mM Tris–HCl [pH7.5], 10mM MgCl2, 0.1% NP-40, 0.1% 2-mercaptoethanol) with or without 20 µM of cyclocytidine HCl for 16 h at 37 °C. NC-associated DNAs were released by 0.5% SDS and 0.5mg/mL PK, and were subsequently subjected to a Southern blot as described above. All experiments were performed at least three times independently.

Statistical analysis

Statistical differences between the mean values were analyzed by the student’s t-test using SPSS software, version 20. A threshold of p < 0.05 was considered statistically significant.

Results

Cytotoxicity of cyclocytidine HCl on HepAD38 cells

After 2d treatment, cytotoxicity of cyclocytidine HCl was evaluated by an MTT cell viability assay, which showed that concentrations over the 20 µM exhibited over 50% cytotoxicity (Fig. 1A). Thus, the concentration of 5, 10, or 20 µM cyclocytidine HCl was applied to subsequent experiments. In addition, cells were treated with 20 µM cyclocytidine HCl and the cell viability was measured at day 2 and day 4. As a result, after 4 days treatment, cells exhibited about 50% of cell viability (Fig. 1B).

Figure 1 Cytotoxicity of cyclocytidine HCl in HepAD38 cells.

(A) HepAD38 cells were treated with different doses of cyclocytidine HCl and the cell viability was measured using a MTT assay. (B) HepAD38 cells were treated with 20 uM of cyclocytidine for 2 days and 4 days, respectively, and the cell viability was measured using a MTT assay.

Cyclocytidine HCl decreased rcDNA production

To study the inhibitory effect of cyclocytidine HCl on HBV replication in dose–response, HepAD38 cells were treated with 5, 10, or 20 µM of cyclocytidine HCl. As a result, 20 µM cyclocytidine HCl appeared to significantly decrease rcDNA production without affecting the nsDNA synthesis as shown by a comparison of the relative ratio of rcDNA to nsDNA (Fig. 2A, Lane 3 and 4 vs. lane 1; Fig. 2B) (p < 0.05). These findings suggest that cyclocytidine HCl may inhibit the synthesis of the positive strand of rcDNA. In addition, cyclocytidine HCl also decreased the production of rcDNA in a time-dependent manner without affecting the nsDNA production (Fig. 2C, lane 1 vs. lane 2), however, the extended time (4 days) of treatment decreased the production of total core DNA that might result from cytotoxicity (Fig. 2C, lane 3; Fig. 1B).

Figure 2 Detection of cytoplasmic HBV core DNA.

(A) HepAD38 cells were treated with 5,10, 20 µM of cyclocytidine HCl for 2 days and 18 µL of the NP-40 lysates were digested with 0.5% SDS and 0.5 mg/mL PK to release the HBV DNA from NC, and viral DNA was detected by a Southern blot. (B and D) The relative ratio of rcDNA to ssDNA was compared between DMSO and the cyclocytidine HCl-treated group. (C) HepAD38 cells were treated with 20 µM cyclocytidine HCl for 2 days and 4 days, respectively. The total amount of core DNA was released by SDS and PK treatment and detected by southern blot. The data was presented as the mean ± SD and the statistical differeces between the DMSO-group and cyclocytidine-group were analyzed by the student’s t-test. A p < 0.05 was considered statistically significant and the * represent p < 0.05. Equal amount of cytosolic nucleic acid in stained gel was reflected the equal amount of sample loading (Fig. S1). Abbreviations: rc, relaxed circular; ns, negative strand; PK, protease K.

Cyclocytidine HCl inhibited the synthesis of the positive strand of rcDNA

To understand whether the reduction of rcDNA was caused by the inhibitory effect of cyclocytidine HCl on the activity of HBV POL, EPR, a cell-free reaction, was performed as described above. As a result, decreased nsDNA and increased rcDNA production indicated that the positive strand DNA was successfully synthesized by HBV POL in EPR without cyclocytidine HCl (Fig. 3A, lane 1 vs. lane 2). However, in presence of 20 µM cyclocytidine HCl, there was no marked increase of rcDNA production (Fig. 3A, lane 1 vs. lane 3). This finding implies that cyclocytidine HCl may block the synthesis of the positive strand of rcDNA by inhibiting HBV POL activity. In addition, the total amount of core DNA was not affected in EPR, suggesting cyclocytidine HCl had no effect on the stability of NC (Fig. 3C).

Figure 3 Synthesis of positive strand of rcDNA by EPR.

(A) After EPR, NC associated DNA was digested with 0.5% SDS and 0.5 mg/mL PK and detected by a Southern blot. (B) The ratio of rcDNA to nsDNA was compared between the DMSO and the cyclocytidine HCl-treated groups. (C) The total amount of core DNA was compared between DMSO and cyclocytidine HCl treated group. The data was presented as the mean ± SD and the statistical differeces between the dNTP(EPR)-group and dNTP(EPR) + cyclocytidine-group were analyzed by the student’s t-test. A p < 0.05 was considered statistically significant and the * represent p < 0.05. Abbreviations: rc, relaxed circular; ns, negative strand; PK, protease K.

Discussion

Synthesis of HBV rcDNA relies on the RT and DNA Pol activity of HBV POL. The activity of RT contributes to synthesis of the negative strand of rcDNA using pgRNA as a template, and DNA Pol activity further extends the positive strand of rcDNA using nsDNA as a template (Hu & Seeger, 2015). In the present study, HepAD38 cells treated with cyclocytidine HCl showed a reduction in rcDNA production without affecting nsDNA synthesis, implying that cyclocytidine HCl could inhibit the synthesis of positive strand of rcDNA. This could have been achieved either by possibly competing with normal deoxycytidine after being hydrolyzed into cytarabine and phosphorylated into the triphosphate form by cellular enzymes for incorporation into DNA synthesis or by directly inhibiting the HBV POL activity during positive strand DNA synthesis, or by both pathways. Intriguingly, the non-phosphorylated form of cyclocytidine HCl also blocked the synthesis of positive strand of rcDNA in a cell-free EPR assay, suggesting that the non-phosphorylated form of cyclocytidine HCl might also directly inhibit HBV POL activity. Moreover, cyclocytidine HCl treatment did not decrease the total amount of core DNA in EPR, indicating cyclocytidine HCl may not destabilize the NC.

It has been thought that rcDNA is the precursor of cccDNA, and theoretically, a reduction in rcDNA production could decrease cccDNA formation in nucleus (Gao & Hu, 2007; Guo et al., 2007). However, the failure of cccDNA detection, possibly due to the small amount of cccDNA and low sensitivity of digoxigenin-dUTP labeled DNA probes, we failed to detect the changes in cccDNA production.

Conclusions

The present study showed that cyclocytidine HCl could inhibit the synthesis of the positive strand of HBV rcDNA either by incorporation into the DNA strand (triphosphorylated form) or inhibit HBV POL activity (non-phosphorylated form). A high-test concentration (20 µM) may not be suitable for anti-HBV treatment, especially since highly efficacious anti-HBV drugs are already available, however, the inhibitory effect of cyclocytidine HCl on HBV POL provides a clue that it could be used for other DNA viruses relying on viral or host DNA polymerase. These findings shed light on usage of an “old” drugs for a “new” application.

Supplemental Information

Supplemental Information 1 Blot of Figure 2A

Click here for additional data file.

Supplemental Information 2 Blot of Figure 3C

Click here for additional data file.

Supplemental Information 3 Blot of Figure 4A

Click here for additional data file.

Supplemental Information 4 Quantification of cytosolic nucleic acid

After finishment of agarose gel electrophoresis, the cytosolic nucleic acid in the gel was visiualized under ultroviolate light and the image was taken by gel imaging system. the densitometry data of nucleic acid was measeured by ImageJ software and exibited as bar chart.

Click here for additional data file.

Supplemental Information 5 Figure 2A (Stained Gel)

Click here for additional data file.

Supplemental Information 6 Figure 2B (Stained Gel)

Click here for additional data file.

Supplemental Information 7 Raw data of MTT result

Click here for additional data file.

Supplemental Information 8 Raw data of HBV DNA quantification of figure 3B

Click here for additional data file.

Supplemental Information 9 Raw data of MTT result for Figure 1B

Click here for additional data file.

Supplemental Information 10 Raw data of HBV DNA quantification of figure 3D

Click here for additional data file.

Supplemental Information 11 Raw data of HBV DNA quantification of Figure 4B and 4C

Click here for additional data file.

Supplemental Information 12 Nucleotide sequence of HBV DNA

Click here for additional data file.

Additional Information and Declarations

Competing Interests

Author Contributions

Data Availability

The authors declare there are no competing interests.

Xue Wang performed the experiments, prepared figures and/or tables, and approved the final draft.

Yihan Xiao performed the experiments, authored or reviewed drafts of the article, and approved the final draft.

Zhigang Cui performed the experiments, authored or reviewed drafts of the article, and approved the final draft.

Zongxin Li analyzed the data, authored or reviewed drafts of the article, and approved the final draft.

Lihua Li analyzed the data, authored or reviewed drafts of the article, and approved the final draft.

Lixian Wu analyzed the data, authored or reviewed drafts of the article, and approved the final draft.

Feifei Yin conceived and designed the experiments, analyzed the data, authored or reviewed drafts of the article, and approved the final draft.

Xiuji Cui conceived and designed the experiments, analyzed the data, prepared figures and/or tables, authored or reviewed drafts of the article, and approved the final draft.

The following information was supplied regarding data availability:

The raw measurements are available in the Supplemental Files.

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
