# Peer review of "Cyclocytidine hydrochloride inhibits the synthesis of relaxed circular DNA of hepatitis B virus"

_PeerJ, doi:10.7717/peerj.13719_

## Round 0.1 · original submission · Major Revisions

Dear authors:

Three experts have revised your manuscript and they all provided positive comments about its content. However, the lack of statistical analysis and several aspects that require your attention, the manuscript needs several improvements. I kindly request that you take the observations done by the three experts, address them, and provide a revised manuscript with a letter replying to all the comments done by the reviewers. Thank you for submitting your work to PeerJ. Best regards, Bernardo

·

Basic reporting

The manuscript addresses an interesting subject, suitable for publication in this journal. There are some concerns about the methodology and results in the presentation that needs to be addressed before providing a positive recommendation.

Experimental design

How was the Southern blot performed? Please add details about these experiments. What was used as a probe? Most importantly, how much DNA was loaded per lane? The figure legends stated that certain volumes were used per lane, but this is incorrect. The lanes should include the same amount of DNA, for comparison purposes.
Please include details about the densitometric analyses applied to southern blot images.
Importantly, there is no statistical analysis applied to results, and this is mandatory to draw conclusions. In the same line, a statistical analysis section should be included in the methodology.

Validity of the findings

Figures 3 and 4. I agree with the results presented as a ratio between both DNA forms, but I also suggest including the relative abundance of RC and NS and comparing them across the samples from cells treated with different concentrations of the compound. Results suggest that NS abundance is also affected by the drug.

Additional comments

Figure 1 should be deleted, adds no value to the manuscript content.

Reviewer 2 ·

Basic reporting

Clear, concise, and well founded.

Experimental design

Simple and clear. Basic toxicity test, followed by measurements of drug inhibition. There could be additional work, but the work contained herein is excellent.

Validity of the findings

I had one major concern with figure 4C (see comments), but it should be easily remedied.

Additional comments

Small grammatical or spelling errors were noted. Otherwise the English and writing were excellent.

Annotated reviews are not available for download in order to protect the identity of reviewers who chose to remain anonymous.

·

Basic reporting

In this paper the authors tested the effect of cyclocytidine HCl on HBV DNA synthesis. They found that the cyclocytidine HCl decreased the synthesis of HBV rcDNA, and they hypothesize than this might be due to the inhibition of viral DNA polymerase activity.

General Comments

This is an interesting, straight forward work, with clear aims and an adequate experimental design. However, a number of observations should be answered by the authors before the manuscript is suited for publication.

A number of grammatical and language style mistakes were found throughout the manuscript. Thus, we suggest a deep review of English language.

The main deficiency of the present work is the lack of statistical analysis of data presented.

Introduction

The authors wrote that “Current anti-HBV agents primarily target the process of rcDNA synthesis, but cannot inhibit or destroy cccDNA formation”. It would be important to describe what is the actual effect of the mentioned anti-HBV agents, and why there is a need to look for new anti-viral agents.

Although cyclocytidine HCL is the agent evaluated in this work, the information provided about it in the Introduction section is very scarce, it is not clear why the authors selected this particular prodrug. Please write a clear justification to study cyclocytidine HCL as a potential anti-viral agent.

Experimental design

Materials and Methods

Brand names of relevant reagents are only cited in a few of them. Please indicate the brand name of the most important reagents included in the present work.

There is not a subsection of Statistical Analysis. To support your findings a basic statistical analysis should be performed, and described in the Materials and Methods section.

Validity of the findings

Results

Sub-section “Cyclocytidine HCl decreased rcDNA production”. The authors declare that “20 µM cyclocytidine HCl appeared to moderately decrease rcDNA production”, “moderately decrease” is inconclusive, the authors must show a statistical analysis to support this observation.

In lines 145-146, the authors describe that “…the extended time (4 days) of treatment decreased the production of total core DNA that might result from cytotoxicity (Figure 3C, lane 3).” It would be interesting to demonstrate that the effect on core DNA production is actually due to cytotoxicity, a simple assay incubating cells with 20 µM cyclocytidine HCl at different time points (including 2 and 4 days) evaluating cell viability with the MTT would provide a clear answer to this point.

Sub-section “Cyclocytidine HCl inhibited the synthesis of the positive strand of rcDNA”. Although the effect on rcDNA synthesis seems to be clear, a statistical analysis is needed to support the findings.

Figures

Figure 1. There is a number of inconsistencies in Figure 1 capture. For instance, the authors indicate “(left) Relaxed circular (rc) DNA.” , however there is no need for indicating “left” since the title “Relaxed circular DNA” is written. Same comment with the “right” section of the Figure. The authors do not declare what “POL” means. Please correct this Figure caption.

Figure 2. Figure caption, the authors declare that “The Y axis represents the relative cell viability, and the X axis represents the dose of cyclocytidine HCl.”, again, there is no need for this phrase, since axis titles are clear enough.

Figure 3. The authors declare that “The data was presented as the mean±SD”, however the authors do not indicate whether the differences shown are significant. Statistical analysis of the results is needed.

Figure 4. The authors declare that “(B) The ratio of rcDNA to ssDNA was compared between the DMSO and the cyclocytidine HCL-treated groups. (C) The total amount of core DNA was compared between DMSO and cyclocytidine HCl treated group.” In the actual figure it is not clear where DMSO treatment was used.

“The data was presented as the mean±SD.” Again, the authors do not indicate whether the differences shown are significant. Statistical analysis of the results is needed.

Conclusions

Although the results showed that cyclocytidine HCl had a potentially favorable anti-viral effect the authors significantly limit the meaning of their findings by declaring that “A high test concentration (20 µM) may not be suitable for anti-HBV treatment, especially since highly efficacious anti-HBV drugs are already available…” then what is the relevance of the work?, what is the “light” that the results “shed on usage of an “old” drugs for a “new” application”, what are the perspectives of the work ?

---

## Round 0.2 · Major Revisions

Dear professor Cui and colleagues:

One of the reviewers is rising a major comment that I totally agree with that is related to the total amount of DNA loaded for Southern analysis. I kindly request including the loading control for this analysis, this can be a stained gel before transfer or other tool as seen fit. Also, as supplementary material, please add the map of the probe used indicating the complementary region. Thank you so much.

Best regards,
Bernardo

·

Basic reporting

The manuscript was improved, but still contains flaws that need to be solved.

Experimental design

My original question about the probe was not addressed. Please include the region coordinates where the probe is hybridizing.
The problem originally spotted with the DNA loading on gels still remains. The authors need to provide Southern blot images generated with the same DNA loaded per lane, otherwise, the comparison between lanes is meaningless.

Validity of the findings

Data are not valid, as southern blots were not generated with the same amount of DNA per lane.

·

Basic reporting

All comments have been properly attended

Experimental design

All comments have been properly attended

Validity of the findings

All comments have been properly attended

Additional comments

All comments have been properly attended

---

## Round 0.3 · accepted · Accept

Dear authors,

I thank you for assessing all the comments that the two experts in the field made to your study. I find the study interesting and with future experiments to elucidate further the effect of cyclocytidine.

Therefore, I thank you for choosing PeerJ for submitting your work.

Warm regards,
Bernardo

·

Basic reporting

The manuscript was improved.

Experimental design

The manuscript was improved.

Validity of the findings

The manuscript was improved.

Additional comments

The manuscript was improved.